# Antimicrobial Profile of Actinomycin D Analogs Secreted by Egyptian Desert *Streptomyces* sp. DH7

**DOI:** 10.3390/antibiotics10101264

**Published:** 2021-10-18

**Authors:** Dina H. Amin, Hayam A. E. Sayed, Ahmed M. Elissawy, Dina E. EL-Ghwas, Abdel Nasser B. Singab

**Affiliations:** 1Department of Microbiology, Faculty of Science, Ain Shams University, Cairo 11566, Egypt; hayam84@sci.asu.edu.eg; 2Department of Pharmacognosy, Faculty of Pharmacy, Ain Shams University, Cairo 11566, Egypt; aelissawy@pharma.asu.edu.eg (A.M.E.); dean@pharma.asu.edu.eg (A.N.B.S.); 3Center of Drug Discovery Research and Development, Faculty of Pharmacy, Ain Shams University, Cairo 11566, Egypt; 4Pharmaceutical Industries Researches Division, Department of Chemistry of Natural and Microbial Products, National Research Centre, Giza 12622, Egypt; dinaelghwas7781@yahoo.com

**Keywords:** desert actinobacteria, MRSA, actinomycin D, Sinai desert, Egypt

## Abstract

Egyptian deserts are an underexplored ecological niche, especially the Sinai Peninsula. In our recent study, we explored this extreme environment and shed light on the bioactive capabilities of desert *Actinobacteria* isolated from Sinai. Fifty desert *Actinobacteria* were isolated from the Sinai desert using mineral salt media, basal media, and starch casein media. The filtrate of *Streptomyces* sp. DH 7 displayed a high inhibitory effect against multidrug-resistant *Staphylococcus aureus* (MRSA) strains. The 16S rDNA sequencing confirmed that isolate DH7 belongs to the genus *Streptomyces*. The NJ phylogenetic tree showed relatedness to the *Streptomyces flavofuscus* strain NRRL B-2594 and *Streptomyces pratensis* strain ch24. The minimum inhibitory concentrations against MRSA were 16 and 32 μg/μL. Chemical investigation of the ethyl acetate extract of *Streptomyces* sp. DH7 led to the isolation and purification of natural products **1**–**4**. Structure elucidation of the purified compounds was performed using detailed spectroscopic analysis including 1 and 2D NMR, and ESI-MS spectrometry. To the best of our knowledge, this is the first report for the isolation of compounds **1**–**4** from a natural source, while synthetic analogs were previously reported in the literature. Compounds **3**–**4** were identified as actinomycin D analogues and this is the first report for the production of actinomycin D analogs from the Sinai desert with an inhibitory effect against MRSA. We indorse further study for this analog that can develop enhanced antimicrobial activities. We confirm that the desert ecosystems in Egypt are rich sources of antibiotic-producing *Actinobacteria*.

## 1. Introduction

Multidrug-resistant pathogens are a life-threatening problem that has affected the healthcare domain over the last few decades. There is a consensus that the data collection processes need to be improved, particularly in Africa [1]. Antimicrobial resistance is related to an increase in morbidity, mortality, hospitalization time, and costs of treatment [2,3]. Multidrug-resistant *Staphylococcus aureus* (MRSA) is among the top antimicrobial resistances responsible for many antibiotic-resistant infections worldwide [1,4]. *Staphylococcus aureus* causes various skin and soft tissue infections, pneumonia, meningitis, bacteremia, osteomyelitis, endocarditis, toxic shock syndrome, and sepsis [5,6,7,8]. Multidrug-resistant *Staphylococcus aureus* has become overwhelmingly popular in Egyptian health care sectors against several antibiotics [9,10]. Over the years, the treatment of suspected *S. aureus* infection has been complicated due to the resistance to multiple drug classes. Exploring novel antibiotics that would combat earlier infection has become vital.

*Actinobacteria* are among the promising sources of alternative antibiotics [11,12,13]. *Streptomyces* is the largest genus of *Actinobacteria* that belongs to the family *Streptomycetaceae* [14]. These gram-positive bacteria are saprotrophic in soil and water. Streptomycetes are a rich source of active secondary metabolites such as clinically useful antibiotics of natural origin [11,15]. The approach of screening less exploited *Actinobacteria* in extreme environments is valuable for the discovery of new chemical entities [16]. The 16S rRNA gene sequence and phylogenetic tree analysis are useful methods for the specific identification of *Actinobacteria* on the genus level and show the differences between the newly identified isolates and other related strains [17,18].

Desert habitats are one of the extremophilic ecosystems that harbor biologically active *Actinobacteria* strains. They are viewed as an ecosystem equivalent to that of the planet Mars [19]. They contain various unexplored *Actinobacteria* that withstand abnormal chemical conditions including pH, salinity, and water quality, and physical conditions including temperature, pressure, and radiations. Egyptian deserts are old sources of undiscovered extremophile and extremotroph *Actinobacteria* strains [20]. The extremophiles grow and flourish at extreme scopes of these physicochemical parameters. In contrast, extremotrophs can grow and tolerate only extreme conditions [21]. The desert of Sinai occupies about 6% of Egypt and is part of the Sahara-Arab desert. It is viewed as one of the low humidity areas with high mountains and occasional snowfalls in extreme natural environments. The extreme habitat of the Sinai desert varies from other places in Egypt, suggesting that the population of *Actinobacteria* is likely to vary, alongside, therefore, their biological abilities [22,23].

In this manuscript, we report the power of screening unusual habitats to find new natural products that attack multidrug-resistant *S. aureus* pathogens.

## 2. Results

### 2.1. Isolation of Desert Actinobacteria Isolates

Fifty *Actinobacteria* were isolated from five soil samples at different parts of the Sinai desert in Egypt. *Actinobacteria* colonies’ growth was detected on mineral salt media, basal media, and starch casein media at 30 °C only. No colonies were observed at 37 °C and 45 °C on the previously stated media types. Starch casein media was the most effective as 50% of *Actinobacteria* isolates were recovered on it.

### 2.2. Antimicrobial Potential of Desert Actinobacteria Isolates

Sixteen desert *Actinobacteria* (32% of total isolates) were found to inhibit one or more of the pathogenic multidrug-resistant *S. aureus* isolates, while 34 (68%) showed no antimicrobial activities on the starch casein broth medium. Antimicrobial activities varied against the tested *S. aureus* isolates and only four isolates (8%) showed an inhibitory effect against *Staphylococcus aureus* ATCC 6538. Fourteen isolates (82%) were inhibitive against multidrug-resistant *Staphylococcus aureus* clinical strains from the Egyptian Hospital. Regarding the specific behavior of the crude extract of isolate DH7 on *S. aureus* cells, it significantly affected cells’ viability of *S. aureus* isolates (No. 35, 37, 39, and 43). The inhibition zone diameters of DH7 crude extract and CN10 antibiotic were 30 mm and 20 mm against *S. aureus* isolates (No. 31) and (55), respectively. The AZM15 antibiotic showed inhibition zone diameters of 26 mm and 28 mm against *S. aureus* isolates (No. 31) and (No. 55), respectively (Table 1). Our findings confirmed that desert *Actinobacteria* showed significant antimicrobial effects against pathogenic multidrug-resistant *S. aureus*. Among them, DH7 reported the highest spectrum of antimicrobial activities against all the tested pathogens. Our preliminary results recorded the marked antimicrobial activities of the DH7 isolate against *S. aureus* ATCC 6538 and against nine multidrug-resistant *S. aureus* isolates.

### 2.3. Morphological Characterization

The cultivation of isolate DH7 on starch casein media for 7 days at 30 °C showed a gray powdery texture, a yellow substrate mycelium, and a yellow exopigment. Microscopic examination using scanning electron microscopy showed the long spore chains of isolate DH7. Spores were globose in shape with a smooth surface (Figure 1).

### 2.4. 16S rRNA Genes Sequence Analysis and Phylogenetic Tree Construction

Amplicons of 500-bp segments of the 16S rRNA gene were effectively amplified from partial 16S rRNA genes of DH7. Purified 500-bp fragments of the 16S rRNA gene were sequenced and the partial 16S rRNA gene sequence was submitted to Gen-bank with the accession number (MN153036).

The 16S rRNA gene sequence of isolate DH7 was compared to the nucleotide sequences of *Streptomyces* strains in the NCBI GenBank database. The comparison showed that *Streptomyces fulvissimus* strain DSM 40,593 was the closest match to isolate DH7 with blast identity (99%). Phylogenetic analysis showed one main clade in which isolate DH7 was amongst other *Streptomyces* strains in the same clade. Similar 16S rRNA gene sequence belonging to *Streptomyces flavofuscus* strain NRRL B-2594 and *Streptomyces pratensis* strain ch24 were found to fall under the same category with bootstrap (86%, Figure 2). Based on previous investigations, we confirmed that strain DH7 isolated from Egyptian soil was identified as belonging to the genus *Streptomyces* in the family *Streptomycetaceae* and was also identified as *Streptomyces* sp. DH7.

### 2.5. Evaluation of the Antimicrobial Activity of the Active Fraction of Streptomyces sp. DH7

The MIC and the MBC of the active fraction (intermediate compound during purification stages) performed using the micro broth dilution method were similar. Our data emphasized that the active fraction produced by *Streptomyces* sp. DH7 was highly inhibitive. *S. aureus* ATCC 6538 was susceptible to the active fraction with MIC of 16 µg/µL. A gradual inhibition occurred at 0.5, 1, 2, 4, and 8 μg/μL, followed by a sharp decline, indicating robust inhibition at the concentration of 16 μg/μL. Complete inhibition of *S. aureus* ATCC 6538 was recorded at concentrations of 32, 64, 128, and 256 μg/μL (Figure 3). Results were confirmed by the failure of *S. aureus* ATCC 6538 to grow on nutrient agar plates of 32, 64, 128, and 256 μg/μL, indicating a lack of viability. The previous procedures also confirmed that the minimum bactericidal concentration was 32 μg/μL, as it was the lowest broth dilution of the active fraction that prevented *S. aureus* ATCC 6538 growth on the agar plate. Four compounds could be isolated from the active fraction. Only compounds **3** and **4** showed a high inhibitory effect against MRSA strains with a 30 and 50 mm diameter of the inhibition zone, respectively (Figure 4 and Figure 5). We concluded that the biological compound produced by *Streptomyces* sp. DH7 isolated from the Egyptian Sinai desert is extremely active against multidrug-resistant *S. aureus*. This is the first study to target clinical multidrug-resistant *S. aureus* using the extracts of desert *Actinobacteria* isolated from the Sinai desert in Egypt.

### 2.6. Structural Determination of the Compounds Isolated from the Active Fraction

Four compounds were isolated from the active fraction of *Streptomyces* sp. DH7. Compound **1** (Figure 6) was isolated as an off-white powder and an APT spectrum of **1** revealed the presence of six carbon signals classified as one methyl, two methylenes, one methine, and two quaternary carbons. Careful inspection of the chemical shift (δ_C_) values confirmed the presence of ketone and amide groups at δ_C_ 205 and 172 ppm, which were assigned as C-2 and C-1′, respectively. The ^1^H NMR spectrum showed the presence of a de-shielded methyl group at δ_H_ 2.01 ppm, assigned as the acetamide group, as well as four diasterotopic methylene protons, assigned as H-3 and H-4 at δ_H_ 3.44 and 3.29 for H-3A and H-3B, in addition to δ_H_ 2.59 and 2.15 for H-4A and H4B. COSY correlations (Figure 7) confirmed the H-H connectivity with strong correlations between H-1 at δ_H_ 4.66 and H-4A and B, whereas the latter showed correlations with H-3A and 3B. Long-range HMBC correlations (Figure 7) showed *J*_3_ and *J*_2_ correlations that confirmed the precise positioning of the functional groups with the Methyl protons at δ_H_ 2.01, which showed only one correlation with the amide carbon C-1′, confirming its assignment as a terminal acetamide methyl group. Moreover, H-1 at δ_H_ 4.66 showed two *J*_3_ correlations with amide carbon and C-3, and one *J*_2_ correlation with ketone carbon, the latter showing one *J*_3_ correlation with H-4A and B as well as one *J*_2_ correlation with H-3A and B, confirming the presence of a cyclobutanone ring. Compound **2** (Figure 6) was isolated as an off-white amorphous powder. The 1D and 2D NMR spectra of 2 revealed a similar pattern to compound **1**, with the disappearance of the terminal acetamide methyl group and the presence of the ethyl group instead, as evidenced by the presence of additional methylene groups at δ_H_ 2.27, showing COSY correlation (Figure 7) with an upfield methyl group at δ_H_ 1.16; both protons (H-2′ and H-3′), in turn, showed strong HMBC correlations (Figure 7) with the amide group C-1′ at δ_C_ 176.1 ppm (Table 2, Appendix A).

Compound **3** (Figure 6) was isolated as a reddish powder and LC-ESI-MS showed pseudo-molecular ion peaks at *m*/*z* 1270 and 636, corresponding to [M+H]^+^ and [M\2 +H]^2+^, respectively, where multiply charged ion peaks are common characteristics in peptide mass spectra. The APT-NMR spectrum of compound **3** showed 63 distinct carbon peaks classified as sixteen methyls, fifteen methines, nine methylenes, and twenty-three quaternary carbons. Among the carbon signals, 12 distinct amide/ester carbons were present in the range δ_C_ 166–173 ppm. Additionally, one carbon at δ_C_ 179 ppm marked the presence of exocyclic conjugated ketone carbon. Moreover, 11 aromatic carbons were distinct in the range δ_C_ 101–147 ppm, including two ortho-coupled protonated carbons at δ_C_ 125.8 and 130.3 ppm. Two aromatic methyls were evident at δ_C_ 7.4 and 15.1 ppm. The values of the carbon chemical shifts together with the long-range HMBC correlations indicated the presence of hexazinone moiety with the conjugated exocyclic ketone at δ_C_ 179 ppm (Table 3).

In addition, the proton NMR revealed typical patterns of the amino acid residues with four distinct doublet NH amide protons at δ_H_ 7.18, 7.75, 8.02, and 8.17 ppm, together with four *N*-Me groups at δ_H_ 2.89, 2.90, 2.92, and 2.96 ppm. Additionally, two distinct de-shielded protons at δ_H_ 6.01 and 6.03 ppm presented on two upfield carbons at δ_C_ 56.34 and 56.42 ppm, marking the presence of two proline residues. The ^1^ H and ^13^ C NMR chemical shifts marking proline residues were matched with the reported chemical shifts of the *cis* conformers [24]. In adding the information from the amide protons, amide methyls, amide/ester carbons, and the *cis*-proline methine protons, we determined the presence of ten amino acid residues with eight amide connections and two ester connections between threonine and valine residues. The additional two amide bonds linked the peptide chains with the phenoxazinone residue through the threonine residue. The proton and APT spectral pattern indicated the duplication of the amino acid residues, i.e., the presence of two sets of five amino acids distributed in two distinct chains.

Long-range HMBC (Figure 7) correlations confirmed the connections of the amino acid residues with key HMBC correlations between *N*-Me groups and both glycine and *N*-Methyl valine residues, together with HMBC correlations between aromatic methyl protons at δ_H_ 2.27 as well as at δ_C_ 179, 113, and 145 ppm, marking its position between an exocyclic conjugated ketone and oxygenated aromatic carbon. Similarly, the other aromatic methyl protons at δ_H_ 2.57 showed correlations with δ_C_ 140.8 and 129.5, and with the protonated aromatic methine at 130.3 ppm. Additionally, COSY (Figure 7) and TOCSY correlations confirmed the presence of isoleucine moiety with distinct correlations between δ_H_ 1.46 (CH2 at δc 25.04) and triplet methyl at δ_H_ 0.91 COSY. TOCSY correlations confirmed the proton spin system of each amino acid residue, including the protonated amide protons. Compound **4** (Figure 6) was isolated as a reddish powder and LC-ESI-MS (Figure 8) showed pseudo-molecular ion peaks at *m*/*z* 1284 and 642, corresponding to [M+H]^+^ and [M\2 +H]^2+^, respectively. The 1D and 2D NMR data of compound **4** showed a similar pattern to compound **3** as shown in Table 4 and Appendix A.

## 3. Discussion

In this study, fifty *Actinobacteria* were isolated from the Sinai desert in Egypt. *Actinobacteria* growth was recorded on mineral salt media, basal media, and starch casein media at 30 °C only rather than at 37 °C and 45 °C. We conducted antimicrobial screening for 50 desert *Actinobacteria* isolated from Egyptian soil against some clinical multidrug-resistant *S. aureus* pathogens. Our results confirmed that 16 out of 50 *Actinobacteria* isolates (32%) inhibited the growth of pathogenic multidrug-resistant *S. aureus.* Similar inhibition zone diameters of the DH7 crude extract and CN10 antibiotic were reported against *S. aureus* isolates No. 31 and No. 55. However, the active fraction produced by *Streptomyces* sp. DH7 showed a higher inhibition zone diameter against *S. aureus* (No. 31) than AZM15 antibiotic. This is an indication of the potency of bioactive compounds produced by *Streptomyces* sp. DH7.

Our results agree with the work of previous reports, declaring the potency of *Actinobacteria* isolated from Egyptian deserts. An Egyptian study reported that 10 isolates out of 75 (0.13% of the active strains) showed inhibition to non-clinical *S. aureus* [25]. It is of interest to mention that our study is unique in showing the effect of desert *Actinobacteria* extracts on *S. aureus* isolates from clinical hospital samples. The results obtained in this study support the fact that nature is the infinite source of potent antibiotics [12,15,25,26]. It is of interest to mention that the phylogenetic analysis of the partial 16S rRNA gene sequencing of *Streptomyces* sp. DH7 showed a low bootstrap value with (78%) *Streptomyces flavofuscus* strain NRRL B-2594, which reflects that *Streptomyces* sp. DH7 is diverse from already known species on the database and hence it has a distinct metabolic profile.

NMR data of compounds **1** and **2** indicate that (Table 3) compound **2** is an ethyl derivative of **1**. HSQC correlations confirmed single bond H-C connections for compounds **1** and **2**. To the best of our knowledge, this is the first report for the isolation of compounds **1** and **2** from a natural source. Regarding compound **3**, 1D and 2D NMR (Table 3) spectra were in accordance with the literature data of the known cyclic peptide actinomycin D with the replacement of one valine residue with isoleucine residue (38). The 1D and 2D NMR data of compound **4** (Table 4) showed a similar pattern of compound **3**; however, HMBC correlations revealed the correlation of isoleucine methylene carbon at δc 25.14 with four methyl protons instead of the two in compound **3**. Moreover, the valine methine carbon at δc 31.8 presented in compound **3**, while it was not present in compound **4**. Additionally, the signal at 25.14 showed high intensity compared to its partner in compound **3,** suggesting its correlation to two methylene carbons instead of one methylene. This result coupled with the mass spectrum suggests that compound **4** is an actinomycin D analog, with the replacement of the two valine residues in both rings with isoleucine residues. Compound **4** was isolated in a lower quantity than compound **3** as some of the carbons in compound **3** were not as clear in the APT spectrum, although they were detected from its HMBC traces. To the best of our knowledge, compounds **3** and **4** are new natural products not isolated before from a natural source, although they have been chemically synthesized [27].

Actinomycin D (AMD) is a peptide antibiotic secreted by *Streptomyces melanochromogenes* and *Streptomyces parvullus* consisting of a planar 2-aminophenoxazin-3-one chromophore and two large cyclic pentapeptide lactones [28]. It has been used in the treatment of extremely aggressive malignancies as an anticancer agent in clinical trials [29]. AMD is being used to treat high-risk cancers in conjunction with other anticancer medicines [30]. AMD binds non-covalently to DNA and strongly inhibits the transcription of DNA to RNA. Consequently, it became a strong tool in biochemistry, molecular, and cell biology research [31]. However, in our study, we reported that the Actinomycin analog is produced from *Streptomyces* sp. DH7. Our results agree with that in Zhang et al. in identifying actinomycin D analogs, including D-valine residues (the second amino acid residue in the cyclic depsipeptide of AMD) and the *N*-methyl-L-valine residues (the fifth amino acid residue in the cyclic depsipeptide of AMD). The authors switched them with D-Phe or L- and D-forms of *N*-methylvalines, *N*-methylisoleucine, *N*-methylleucine, *N*-methylphenylalanine, *N*-methylalanine, and sarcosine [27]. However, the analogs identified in our study showed antimicrobial effects against MRSA, which was not previously reported for the actinomycin D analogs identified in the mentioned study [27]. Another study revealed that *Streptomyces griseoruber* NBRC 12,873 isolated from medicinal plants roots, namely actinomycin-D (act-D), presented with inhibitory activity against various gram-positive and gram-negative bacterial cultures [32]. Another research group found that *Streptomyces heliomycini* isolated from the marine coast exhibited strong antibacterial activity against *Staphylococcus aureus*. Using MS and NMR techniques, the active compounds were identified as actinomycins X0β, X2, and D [33]. In 2019, a similar report declared the production of actinomycin D by a genus *Streptomyces* isolated from the Saharan desert [34]. Literature ensures that *Actinobacteria* in different environments produce Actinomycin D, although in our study, we identified Actinomycin analogs with a high inhibitory effect against MRSA strains produced by desert *Actinobacteria*. In 2018, endophytic actinomycetes isolated from the macro fungus *Ganoderma applanatum* were reported to produce 2-methyl-actinomycin D analog, which has better anti-tumor activity than Actinomycin D. Another study declared the identification of novel methylated actinomycin D (mAct D) [35].

Our findings strongly suggest the production of cyclic peptide actinomycin D and actinomycin analogs as produced by desert *Actinobacteria* isolated from the Sinai Peninsula, Egypt. To the best of our knowledge, this is the first report declaring the production of actinomycin analogs inhibiting clinical multidrug-resistant *S. aureus* samples from Egyptian desert *Actinobacteria*. Future studies involving large-scale fermentations and purifications of *Streptomyces* sp. DH7 will be conducted to categorize the properties of these new analogs. For instance, it may prove better antibacterial effects than that of already identified similar ones. These conclusions would overwhelm the resistance mechanisms in the pathogenic bacteria in this area. Our findings will have a positive effect on health research areas and will aid in finding natural solutions to repair fragile ecosystems on both a human and global scale.

## 4. Materials and Methods

### 4.1. Sample Collection

In 2018, five samples were collected from the Sinai desert at Saint Catherine City, south of the Sinai Governorate, Egypt, at GPS coordinates 28°33′42.88″ N 33°56′57.62″ E (Figure 9). Each sample was taken from the desert at 5 cm depth in clean plastic bags. Samples were stored in sterile containers in the refrigerator at 4 °C for further analysis.

### 4.2. Isolation of Actinobacteria from Sinai Desert Soil

Five soil samples were subjected to the serial dilution method. An aliquot of 0.1 mL inoculum of 10^−3^ and 10^−4^ dilution was cultured on mineral salt media, basal media, and starch casein media for 14 days at 30 °C, 37 °C, and 45 °C. Then, colonies were selectively picked and purified on starch casein agar plates for further analysis.

### 4.3. Agar Well Diffusion Method

A preliminary screening of *Actinobacteria* extracts was conducted against 9 multidrug-resistant *S. aureus* strains from a national hospital in Cairo, Egypt, which were previously identified using MALDI (matrix-assisted laser desorption ionization/time-of-flight), and MIC were recorded against several antibiotics (Appendix A). Antimicrobial activity was evaluated using the agar well diffusion method [36]. *Actinobacteria* extracts were performed using a loopful of *Actinobacteria* spores inoculated to 35 mL of starch casein broth media. Flasks were incubated for 7 days at 30 °C in a shaking incubator. Cell-free supernatant of *Actinobacteria* (250 µL) was added in each well in nutrient agar Petri dishes containing 0.5 Mcfarland of MRSA-tested bacterial spores [37]. Petri dishes were then incubated for 24 h at 37 °C and the inhibition zone areas were recorded [38]. Two antibiotics, namely Gentamicin (CN10) and Azithromycin (AZM15), were used as a control in this experiment. All tests and experiments were made in duplicates. Tested MRSA bacteria were cultivated overnight in nutrient broth at 37 °C before the test. A control multidrug-sensitive *Staphylococcus*
*aureus* ATCC 6538 strain was obtained from the Microbial Resources Center (MIRCEN) at the Faculty of Agriculture, Ain Shams University, Cairo, Egypt.

### 4.4. Microscopic Analysis of Isolate DH7

Microscopic characterization of the *Actinobacteria* isolate was examined according to Bergey’s Manual of Systematic Bacteriology, Vol. 4 [39]. Morphological characteristics including aerial and substrate mycelium as well as exopigment production were examined by eyes alone after 8 days of culturing isolate DH7 [40]. The potent *Actinobacteria* isolate DH7 was subjected to further study using a scanning electron microscope. Isolate DH7 was inoculated on starch casein agar and incubated for 7 days, after which a block was then cut from the agar plate and fixed in 2% glutaraldehyde vapor at 37 °C for 3 h. The dehydration step was conducted using a series of ethanol solutions (50, 60, 70, 80, and 95%, 15 min each; 2 times with 100% ethanol, 30 min/time). Ethanol was replaced with acetone, which was subjected to a critical-point dryer, and then coated with gold using a Gold Sputter. Examination by scanning electron microscopy at the Regional Center for Mycology and Biotechnology (RCMB), Al Azhar University, Egypt, was conducted [41].

### 4.5. Extraction of Genomic DNA from Actinobacteria Isolate DH7

DNA extraction was conducted using the GeneJet genomic DNA purification Kit (Thermo K0721). DH7 was inoculated in 35 mL of starch casein broth for 7 days in a shaking incubator at 30 °C. Cells were harvested up to 2 × 10^9^ bacterial cells in a 2 mL microcentrifuge tube, were centrifuged for 10 min at 5000× *g*, and then the supernatant was discarded. The pellet was resuspended in 180 μL of Digestion Solution and an aliquot of 20 μL Proteinase K Solution was added. The sample was mixed and incubated at 56 °C for 30 min. An amount of 20 μL of RNase A solution was added to the cells, mixed, and incubated for 10 min at 37 °C. An aliquot of 200 μL Lysis Solution was added to the sample and mixed for about 15 s. An amount of 400 μL of 50% ethanol was added and mixed by pipetting. The lysate was transferred to a GeneJET™ Genomic DNA Purification Column which was placed in a collection tube and centrifuged for 1 min at 6000× *g*. The GeneJET™ DNA purification column was inserted into a clean 2 mL collection tube and 500 μL of Wash Buffer was added to the sample, followed by centrifugation for 1 min at 8000× *g*. The flow-through was discarded and the purification column was inserted again into the collection tube, followed by the addition of 500 μL of Wash Buffer II to the GeneJET™ DNA purification column. It was centrifuged for 3 min at 12,000× *g*. The collection tube containing the flow-through solution was discarded and the DNA GeneJET™ purification column was transferred to a sterile 1.5 mL microcentrifuge tube. Finally, an aliquot of 80 μL of Elution Buffer was added to the center of the GeneJET™ DNA purification column to elute the genomic DNA. Further incubation for 2 min at room temperature and centrifugation for 1 min at 8000× *g* was performed. The purification column was discarded and the purified DNA concentration was verified using a Nanodrop spectrophotometer (ND-2. 1000, Nanodrop Technologies, Wilmington, DE, USA), after which it was stored at −20 °C.

### 4.6. Polymerase Chain Reaction of 16S-rRNA Genes

The extracted DNA was subjected to PCR analysis of the 16S rRNA gene as described by [42]. Universal bacterial 16S rDNA used included the forward primer (A GTT TGA TCC TGG CTC AG) and reverse primer (GGT TAC CTT GTT ACG ACT T). PCR reaction was performed as follows: 25 μL of the Maxima^®^ Hot Start PCR Master Mix; 1 μL of 20 μM 16SrRNA forward primer; 1 μL of 20 μM 16SrRNA reverse primer; 5 μL of template DNA; and 18 μL nuclease-free water. Amplification was conducted using a thermal cycler (Applied biosystem 337) with initial denaturation at 94 °C for 10 min and then 35 cycles of 30 s at 95 °C, 1 min at 65 °C, 1.30 min at 72 °C, and finally 10 min at 72 °C. The negative control contained all components of the mixture, except the DNA template.

### 4.7. Agar Electrophoresis and PCR Product Purification

PCR products were subjected to gel electrophoresis. One gram of agarose gel was dissolved in 100 mL of TAE buffer and 15 μL of ethidium bromide solution (10 mg/mL Sigma). Agarose gel was poured into the gel casting tray and combs were added. It was cooled until the agarose was solidified. Gel combs were taken off to produce wells for loading. An aliquot of 8 μL of the PCR product was loaded into the wells. A DNA marker (100 bp plus, Vivantis) was loaded into the first well to check the size of the amplified DNA. The gel electrophoresis apparatus was adjusted at 130 V for 40 min. DNA bands were visualized using a gel documentation system (Syngene-Ingenius3). The DNA bands with a size of 500 bp indicated the precise amplification.

The PCR product was cleaned using the GeneJET™ PCR Purification Kit (Thermo K0701). An amount of 45 μL of Binding Buffer was transferred to the PCR mixture and mixed carefully. The mixture was transferred to the GeneJET™ purification column, centrifuged for 30 s at 12,000× *g*, and the flow-through was discarded. Further addition of 100 μL of wash buffer to the GeneJET™ purification column and centrifugation for 30 s at 12,000× *g* were performed. The flow-through was discarded and the purification column was added back into the collection tube. Centrifugation of the empty GeneJET™ purification column was performed for an extra 1 min. The GeneJET™ purification column was added to a clean 1.5 mL microcentrifuge tube with 25 μL of Elution Buffer, followed by centrifugation for 1 min. The GeneJET™ purification column was then discarded and the purified PCR product was stored at −20 °C.

### 4.8. 16S rDNA Sequencing and Phylogenetic Analysis

The PCR product was subjected to Sanger sequencing technology using the ABI 3730xl DNA sequencer at GATC Company, Germany. The 16S rDNA sequences were deposited at the GenBank database (http://blast.ncbi.nlm.nih.gov/, accessed on 1 January 2021) to show the neighboring matches of known species to the 16S rDNA sequence of isolate DH7. Phylogenetic analysis was processed using the CLUSTAL W program [43] and a neighbor-joining phylogenetic tree was constructed using Molecular Evolutionary Genetics Analysis (MEGA) software version 6 [44].

### 4.9. Determination of Minimum Inhibitory Concentrations of the Active Fraction of Streptomyces sp. DH7

The minimum inhibitory concentration (MIC) of the active fraction (an intermediate stage of the purification procedure) was investigated via micro broth assay [36]. A stock solution of the active fraction dissolved in an equal amount of DMSO and was used to prepare different concentrations (0.5, 1, 2, 4, 8, 16, 32, 64, 128, and 256 μg/μL). An aliquot of 5 μL from an overnight culture of *S. aureus* ATCC 6538 (0.5Mcfarland) was added to each well of a sterile 96-well microtiter plate containing the test concentrations of the active fraction. The final volume in each well was 100 μL and each sample was prepared in duplicates. Growth control (bacterial suspension only) and background control (media only) were included in the microtiter plate. The plate was incubated in a static incubator for 24 h at 37 °C. Then, optical densities were recorded using a multi-detection microplate reader (Bio-Tek-Synergy HT Microplate Reader, BioTek Instruments, Winooski, VT, USA) at 600 nm. Further confirmation of MIC was also estimated by transferring 10 µL of the mixture to nutrient agar plates. Plates were then incubated in the static incubator for 24 h at 37 °C. The viability of the bacteria was detected at this stage by an unaided eye. Lack of bacterial growth on the plates implied that only non-viable organisms were present. The kill-growth curve, which shows changes in the optical density of bacterial growth against active fraction concentrations, was plotted using Origin Pro8 for data analysis and graphing (https://www.originlab.com/. Accesed on: 1 December 2020). The MIC is well defined as the lowest concentration of the antimicrobial agent that inhibits visible growth of the tested isolate in nutrient broth was observed with an unassisted eye [36]. Additionally, it is also defined from a spectrophotometric view as the concentration at which there is a sharp decline in the absorbance value.

### 4.10. Determination of the Minimum Bactericidal Concentrations of the Active Fraction of Streptomyces sp. DH7

Minimum bactericidal concentrations (MBC) are defined as the lowest broth dilution of antimicrobials that prevents the growth of the organism on the agar plate [45]. Viability of bacteria in wells of no growth in previously prepared 96-well microtiter plate was checked by transferring 5 µL of the mixture to nutrient agar plates. Then, plates were incubated in a static incubator for 24 h at 37 °C. The bacteria growth survival was detected at this stage using an unassisted eye. No growth indicated the MBC of the active fraction against *S. aureus* ATCC 6538.

### 4.11. Fractionation of Streptomyces sp. DH7 Extract and Bioassay of the Identified Compounds

Spore suspensions of *Streptomyces* sp. DH7 were added to 1500 mL soya bean meal broth media at a ratio of 1% *v*/*v* and were incubated for 8 days at 30 °C (150 RPM). The cell-free supernatant was filtered and tested against *S. aureus* via the agar well diffusion method as previously stated. Bioactive components were extracted from the cell-free supernatant three times using ethyl acetate (1:1 *v*/*v*) in a separating funnel and the ethyl acetate wells were used as the control. The ethyl acetate extract of the fermentation broth was evaporated and the residue was then defatted by partitioning using n-hexane and 90% aqueous methanol.

### 4.12. Purification and Identification of Bioactive Fractions of Streptomyces sp. DH7

The 90% methanolic fraction (600 mg) was applied on vacuum liquid chromatography (VLC) assembly packed with normal-phase silica gel 60 and was eluted using n-hexane (ethyl acetate gradient, from 100% to 0%) followed by dichloromethane (methanol, from 100% to 0%). Then, the fractions were collected, analyzed using TLC with different mobile phases, and similar fractions were pooled together to finally yield 5 major fractions (F1–5).

Fraction 3 eluted with 50% EtOAc in n-hexane (130 mg) was further purified using Sephadex LH-20 eluted with DCM:MeOH (1:1) to yield two major sub-fractions. Sub-fraction 2 (90 mg) yielded a reddish residue and was further analyzed by LC/ESI/MS to reveal a group of peptides. This fraction was further purified using semi-preparative HPLC with 70% Acetonitirile in water as the mobile phase to yield compound 3 (10 mg) and 4 (5 mg).

Similarly pooled fraction 4 eluted with 80% DCM in MeOH (80 mg) was purified using Sephadex LH-20 eluted with MeOH, yielding three sub-fractions. Sub-fraction 3 (40 mg), showing two major spots on TLC, was purified using semi-preparative HPLC with 40% Acetonitirile in water as the mobile phase to yield two pure compounds, namely compound **1** (2 mg) and compound **2** (3 mg).

Normal-phase column chromatography was performed using silica gel 60 (0.04–0.063, Merck, Darmstadt, Germany) on VLC assembly that was equipped with a vacuubrand^®^ (Germany) vacuum pump. TLC analysis was performed using normal-phase silica gel pre-coated plates F254 (Merck, Germany). Size exclusion chromatography was performed using Sephadex LH-20 (Sigma Aldrich, Taufkirchen, Germany).

UPLC/ESI/MS was performed using Shimadzu LCMS 8045, utilizing the UPLC shimpack RP-C18 column (particle size: 3 mm × 70 mm, 2.7 µm) using isocratic elution with 80% acetonitrile in 0.1% formic acid (all solvents are mass grade Sigma Aldrich). Final purification steps were conducted using Shimadzu-LC-20AP preparative HPLC equipped with PDA and the kromasil RP-C18 column (10 mm × 250 mm, 5 µm particle size) detection was done on 235 nm, 254 nm and 280 nm. NMR spectra were recorded using Bruker Avance HD III 400 MHz (switzerland); pure compounds were dissolved in 0.6 mL of deuterated solvents (Sigma Aldrich) and were placed in 5 mm NMR tubes; and proton NMR spectra were recorded at 400 MHz and 13 C NMR at 100 MHz at the Center of Drug Discovery Research and Development, Department of Pharmacognosy, Faculty of Pharmacy, Ain Shams University, Egypt.

### 4.13. Bioassay of the Active Compounds of Streptomyces sp. DH7

The antimicrobial activity against MRSA strains of the four purified compounds was tested as follows: each purified compound was dissolved in DMSO at a concentration of 1 μg/μL. Then, a volume of 20 μL of both compound **3** and **4** was added to sterile filter paper discs and a control disc containing DMSO was also prepared. Using sterile forceps, each disc was added on nutrient agar Petri dishes containing 0.5 Mcfarland of fresh MRSA-tested bacterial cells. Additionally, the agar well diffusion method was performed to record the inhibitory effect for some samples. Petri dishes were then incubated for 24 h at 37 °C and the inhibition zone areas were recorded. All tests were performed in duplicates.

## 5. Conclusions

This study reported the potency of desert *Actinobacteria* isolated from the Sinai desert against clinical multidrug-resistant *S. aureus* strains. We still believe that listening to nature and learning how to communicate with the environment will provide us with biologically safe opportunities to improve human health. This study shows great promise not only for Egypt but also on the global scale. This study highlights actinomycin D analogs isolated from desert *Actinobacteria* as suitable candidates for the treatment of infections caused by multidrug-resistant *S. aureus* pathogens. We suggest that current large-scale fermentations and purifications be performed to further explore the properties of these antimicrobials. We ensure that this study will have a positive impact on human health and social welfare. It will support the medical, pharmaceutical, and public health sectors in Egypt and worldwide.

## Figures and Tables

**Figure 1 antibiotics-10-01264-f001:**
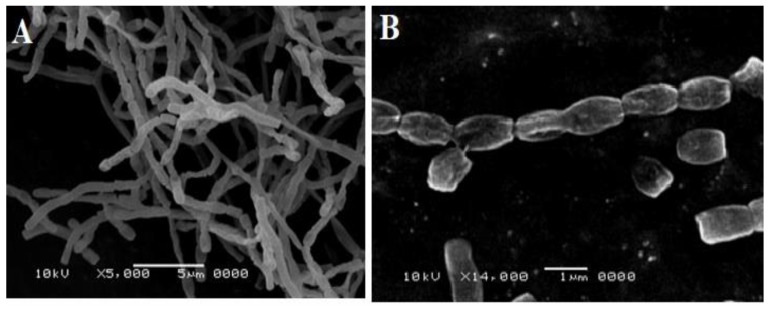
Microscopic examination of isolate DH7 using electron microscopy. Isolate DH7 was cultivated on starch casein media for 7 days at 30 °C. Aerial, substrate mycelium, and spores were examined using a scanning electron microscope: (**A**) magnification of the spore chain (×5000) and (**B**) magnification of the spores (×14,000).

**Figure 2 antibiotics-10-01264-f002:**
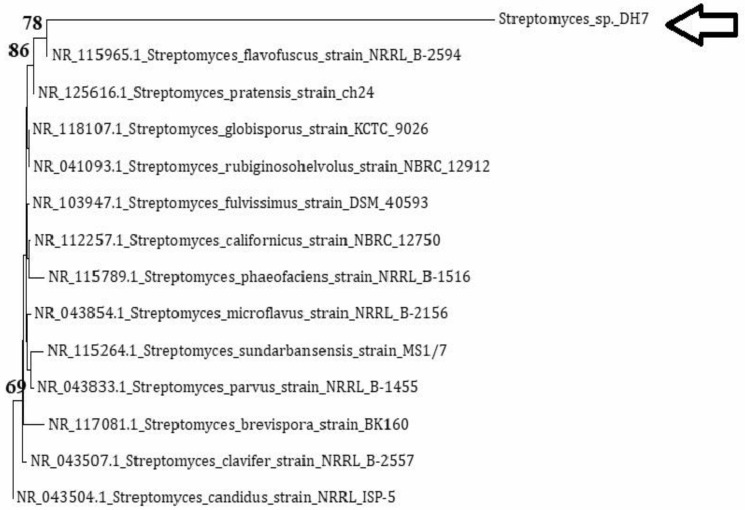
NJ-phylogenetic tree based on partial 16S rRNA gene sequences of *Streptomyces* sp. DH7 and other *Streptomyces* species. Phylogenetic analysis was conducted using the CLUSTAL W program and the neighbor-joining phylogenetic tree was constructed using Molecular Evolutionary Genetics Analysis (MEGA) software version 6.

**Figure 3 antibiotics-10-01264-f003:**
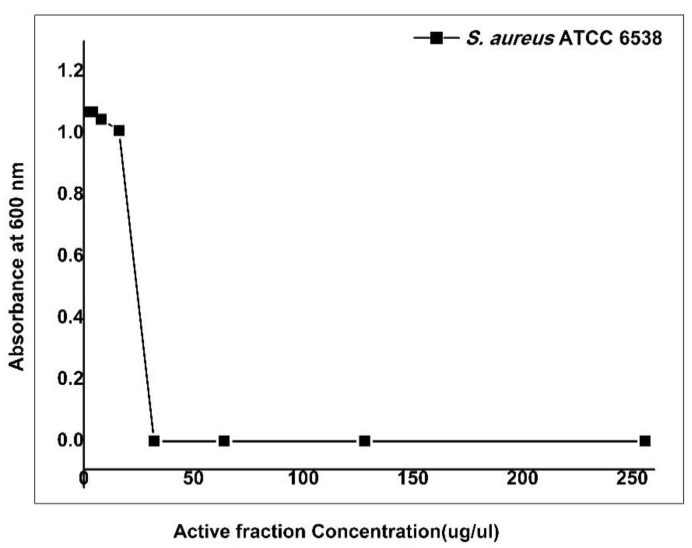
The minimum inhibitory concentration (MIC) of the active fraction of *Streptomyces* sp. DH7 was tested against *S. aureus* ATCC 6538. A stock solution of the active fraction dissolved in an equal amount of DMSO was used to prepare different concentrations, namely 0.5, 1, 2, 4, 8, 16, 32, 64, 128, and 256 μg/μL. An aliquot of 5 μL from an overnight culture of *S. aureus* ATCC 6538 (0.5 Mcfarland) was added to each well of sterile 96-well flat-bottomed microtiter plates containing the test concentrations of the active fraction. Then, optical densities were recorded using a multi-detection microplate reader (Bio-Tek-Synergy HT Microplate Reader, BioTek Instruments, Winooski, VT, USA) at 600 nm.

**Figure 4 antibiotics-10-01264-f004:**
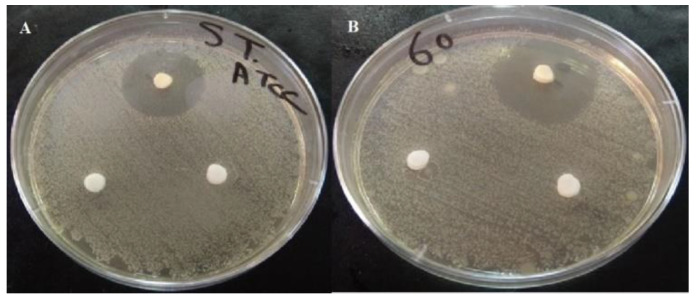
The inhibitive activity of compound 3 isolated from *Streptomyces* sp. DH7 against (**A**) *S. aureus* ATCC 6538 and (**B**) multidrug-resistant *S. aureus* No. 60 via the agar disc-diffusion method. Petri dishes were then incubated for 24 h at 37 °C and then the inhibition zone areas were recorded. All tests were performed in duplicates.

**Figure 5 antibiotics-10-01264-f005:**
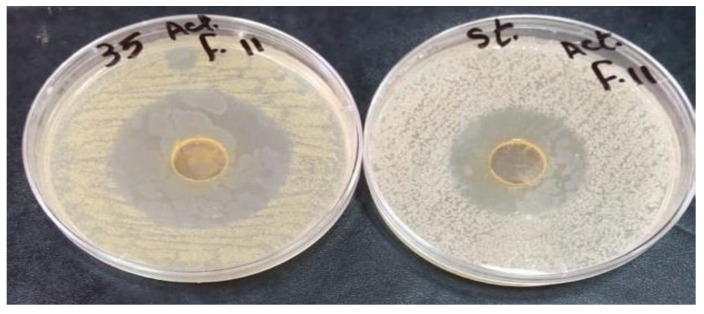
The inhibitive activity of compound 4 isolated from *Streptomyces* sp. DH7 against multidrug-resistant *S. aureus* No. 35 and *S. aureus* ATCC 6538 via the agar well diffusion method. Petri dishes were then incubated for 24 h at 37 °C and then the inhibition zone areas were recorded. All tests were performed in duplicates.

**Figure 6 antibiotics-10-01264-f006:**
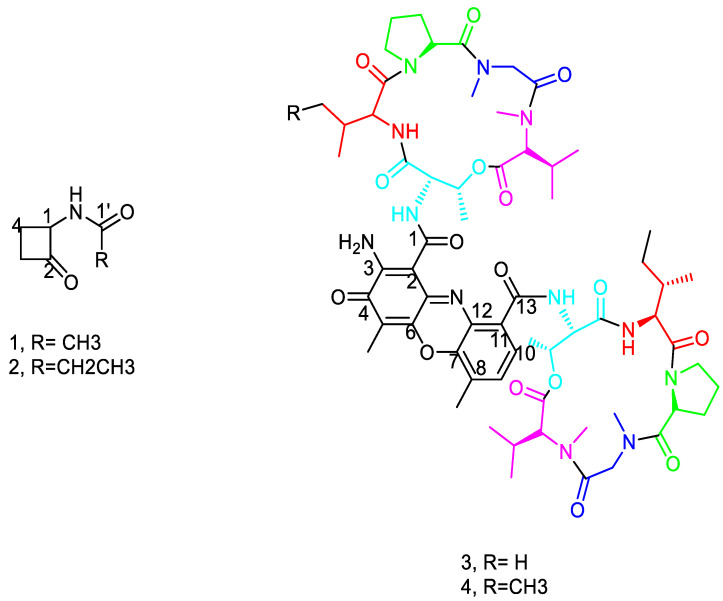
Chemical structures of compounds **1**–**4**.

**Figure 7 antibiotics-10-01264-f007:**
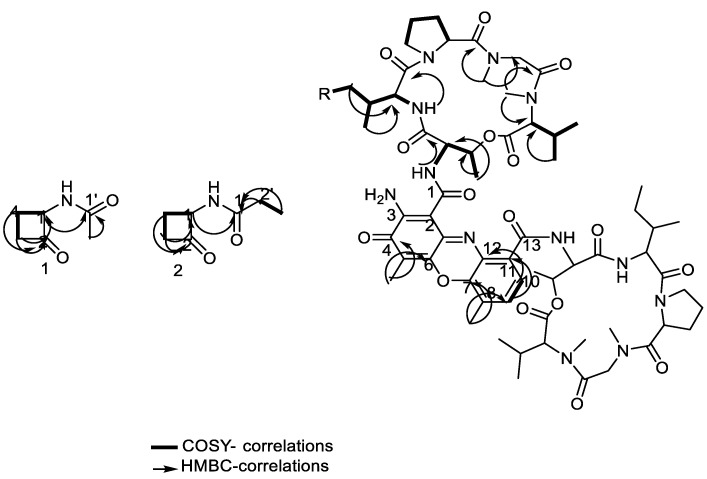
COSY and HMBC correlations of compounds **1**–**4**.

**Figure 8 antibiotics-10-01264-f008:**
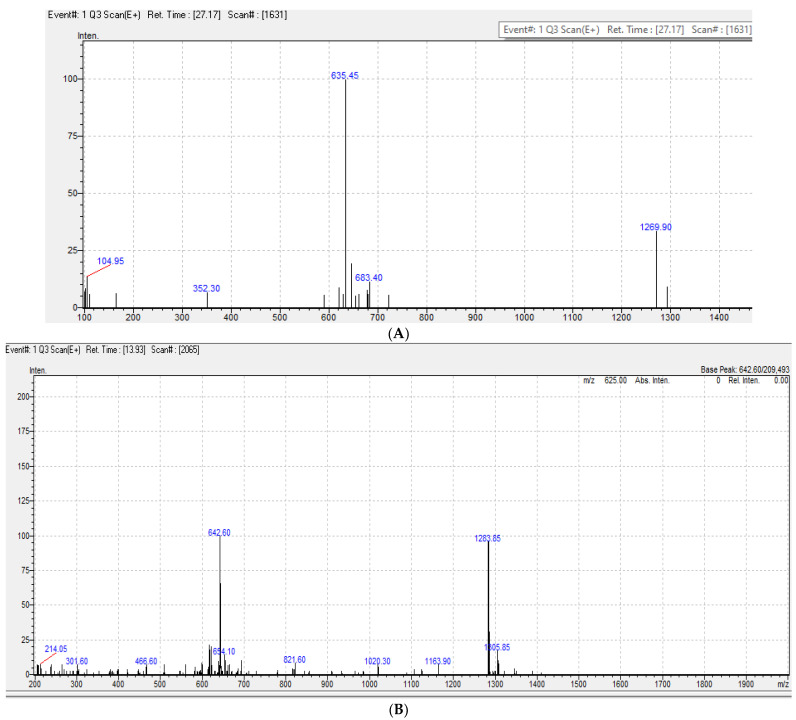
ESI-MS data of compound **3** (**A**) and **4** (**B**).

**Figure 9 antibiotics-10-01264-f009:**
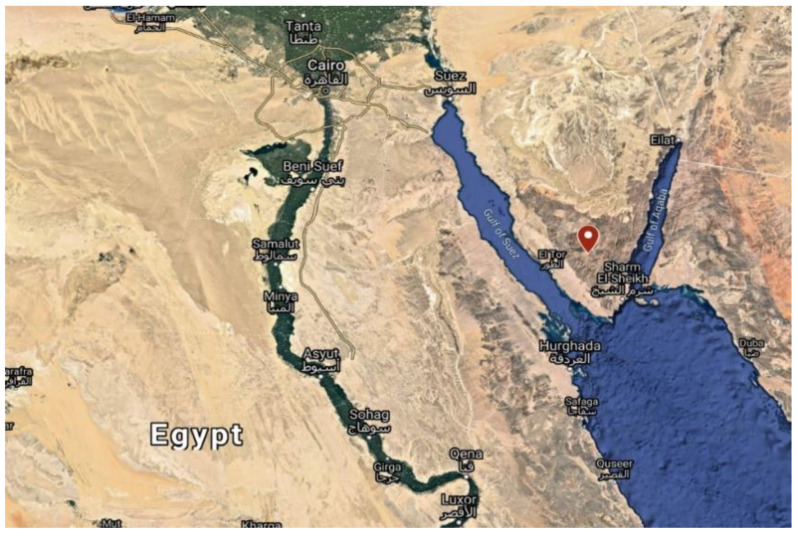
Geography of sampling sites for Actinobacteria isolation from the Sinai desert, south of the Sinai Governorate, Egypt (red spot). This map is retrieved from the satellite map of Egypt (https://www.google.com/maps/. Accessed on: 1 January 2021).

**Table 1 antibiotics-10-01264-t001:** The antimicrobial potential of *Actinobacteria* isolates against multidrug-resistant *S. aureus* strains.

The Diameter of Inhibition Zones in mm against Multidrug-Resistant *S. aureus* Strains										
Tested Isolate	ATCC	No. 30	No. 31	No. 34	No. 35	No. 37	No. 39	No. 43	No. 55	No. 60
DH1	4	-	-	-	10	-	4	-	-	4
DH2	4	-	-	-	-	-	-	-	-	-
DH3	18	-	-	-	-	-	-	-	-	-
DH4	-	-	-	-	-	-	-	12	-	-
DH5	-	12	-	-	-	-	-	-	-	-
DH6	-	-	-	-	-	-	-	-	-	-
DH7	15	4	4	6	4	22	22	6	18	22
DH8	-	-	10	8	4	10	-	10	8	8
DH9	-	6	4	6	-	2	4	-	-	4
DH10	-	-	-	-	-	-	10	-	-	-
DH11	-	-	-	-	-	-	-	-	-	12
DH12	-	12	12	8	16	10	10	-	-	-
DH13	-	-	-	-	-	-	-	-	-	-
DH14	-	-	-	-	-	14	-	-	6	-
DH15	-	-	-	-	-	-	-	-	-	-
DH16	-	-	-	-	-	-	-	-	-	-
DH17	-	-	-	-	-	-	-	-	-	-
DH18	-	12	10	14	10	10	10	16	6	10
DH19	-	-	-	-	-	-	-	-	-	-
DH20	-	-	-	4	-	-	-	-	-	4
DH21	-	20	20	2	-	-	-	2	-	-
DH22	-	-	-	2	-	-	2	6	-	-
DH23	-	-	-	-	-	-	-	-	-	-
DH24	-	-	-	-	-	-	-	-	-	-
DH25	-	-	-	-	-	-	-	-	-	-
DH26-DH50	-	-	-	-	-	-	-	-	-	-
Crude extract DH7	-	-	30	-	32	24	24	24	20	-
Gentamicin (CN10)	-	-	30	-	-	-	-	-	20	-
Azithromycin (AZM15)	-	-	26	-	-	-	-	-	28	-

(-): no inhibition is recorded.

**Table 2 antibiotics-10-01264-t002:** One dimension NMR data of compounds **1** and **2**.

No. of (C-Atoms)	Compound 1		Compound 2	
#	δ_H_ *	δ_C_ *	δ_H_ *	δ_C_ *
1	4.66 (dd, 7,12)	58.3 CH	4.65 (dd, 7,12)	58.5 CH
2	-	205.4 C	-	205.7
3	3.44 (m) 3.29 (m)	27 CH2	3.43 (m) 3.29 (m)	26.6 CH2
4	2.59 (m) 2.15 (m)	30.2 CH2	2.58 (m) 2.14 (m)	30.7 CH2
1′	-	172.2 C	-	176.1 C
2′	2.01 (s)	21.2 CH3	2.27 (*q*, 7.6)	28.6 CH2
3′	-	-	1.16 (*t,* 7.6)	9 CH3
* δ_H_ (CD_3_OD, 400 MHz, *J* in Hz), δ_C_ (CD_3_OD, 100 MHz)				

**Table 3 antibiotics-10-01264-t003:** One dimension NMR data of compound **3**.

Residue *	δ_H_ (CDCl_3_, 400 MHz, and *J* in Hz)	δ_C_ (CDCl_3_ and 100 MHz)	Residue	δ_H_ (CD_3_OD, 400 MHz, and *J* in Hz)	δ_C_ (CDCl_3_ and 100 MHz)
Ring A			Ring B		
Thr	7.75 NH (d, 6.5)	-	Thr	7.18 NH (d, 6.8)	-
	4.62 (dd, 2.3,6.5)	54.9 CH		4.54 (dd, 2.3,6.8)	55.2 CH
	5.22 (m)	75.1 CH		5.22 (m)	75.1 CH
	1.27 (m)	17.5 CH3		1.29 (m)	17.4 CH3
	-	168.5		-	167.5
*N*-Me Val	2.92 (s)	39.2 *N*-Me	*N*-Me Val	2.96 (*N*-Me)	39.3 *N*-Me
	2.7 (m)	71.2 CH		2.8 (m)	71.4 CH
	2.69 (m)	26.9 CH		2.68 (m)	26.9 CH
	0.75 (m)	19.07 CH3		0.76 (m)	19.13 CH3
	0.98 (m)	21.6 CH3		0.99 (m)	21.7 CH3
	-	167.7 C			166.5 C
*N*-Me Gly	2.9 (s)	34.8 (*N*-Me)	*N*-Me Gly	2.91 (s)	34.9 (*N*-Me)
	4.78, 3.63 (m)	51.4 CH2		4.77, 3.65	51.4 CH2
		166.5 C			167.5 C
Pro	6 (d, 9.25)	56.4 CH	Pro	6.03 (d, 9.3)	56.2 CH
	2.11, 2.30 (m)	22.8 CH2		2.13, 2.32	23.0 CH2
	3.75, 3.98 (m)	47.3 CH2		3.75, 3.86	47.6 CH2
	1.85, 2.71 (m)	31.0 CH2		1.87, 2.95	31.2 CH2
		173.2 C			173.3 C
_D-_Val	8.17 NH (d, 6)	-	Isoleu	8.01 NH (d, 6.2)	-
	3.57 (m)	58.9 CH		3.62	58.6 CH
	2.24	31.6 CH		1.91	38.6 CH
	1.15 (d, 6.6)	19 CH3		1.46, 0.96	25.01 CH2
	0.92	19.3 CH3		1.11 (d, 6.6)	14.9 CH3
				0.91 m	12.5 CH3
	-	173.2 C		-	173.7 C
Phenoxazinone					
1		168.8 C			
2		101.8 C			
3		147.5 C			
4		179.1 C			
5		113.5 C			
6		145.2 C			
7		140.5 C			
8		127.9 C			
9	7.38 (d, 7.7)	130.4 CH			
10	7.66 (d, 7.7)	125.9 CH			
11		132.7 C			
12		129.4 C			
13		166.1 C			

* All the amino acid residues are in the L- configuration except for Valine.

**Table 4 antibiotics-10-01264-t004:** One dimension NMR data of compound **4**.

Residue	δ_H_ (CDCl_3_, 400 MHz, and *J* in Hz)	δ_C_ (CD_3_OD and 100 MHz)	Residue	δ_H_(CD_3_OD, 400 MHz, and *J* in Hz)	δ_C_ (CDCl_3_ and 100 MHz)
Ring A			Ring B		
Thr	7.73 NH (d, 6.6)	-	Thr	7.18 NH (d, 6.9)	-
	4.66 (m)	54.9 CH		4.55 (m)	55.2 CH
	5.23 (m)	75.1 CH		5.23 (m)	75.1 CH
	1.28 (m)	17.8 CH3		1.29 (m)	17.4 CH3
	-	168.9		-	166.3
*N*-Me Val	2.93 (s)	39.2 *N*-Me	*N*-Me Val	2.95 (*N*-Me)	39.3 *N*-Me
	2.7 (m)	71.2 CH		2.7 (m)	71.4 CH
	2.68 (m)	26.90 CH		2.68 (m)	26.94 CH
	0.75 (m)	19.07 CH3		0.76 (m)	19.11 CH3
	0.98 (m)	21.6 CH3		0.99 (m)	21.7 CH3
	-	168.5 C			166.2 C
*N*-Me Gly	2.89 (s)	34.9 (*N*-Me)	*N*-Me Gly	2.89 (s)	35.0 (*N*-Me)
	4.78, 3.63 (m)	51.4 CH2		4.77, 3.65	51.4 CH2
		166.4 C			166.5 C
Pro	6 (d, 9.29)	56.5 CH	Pro	6.07 (d, 9.43)	56.2 CH
	2.11, 2.30 (m)	22.8 CH2		2.13, 2.32	23.1 CH2
	3.75, 3.98 (m)	47.3 CH2		3.75, 3.86	47.6 CH2
	1.85, 2.71 (m)	31.0 CH2		1.87, 2.65	31.2 CH2
		173.2 C			173.3 C
Isoleu	8.22 NH (d, 6.0)	-	Isoleu	8.01 NH (d, 6.0)	-
	3.60 (m)	58.4 CH		3.63 (m)	58.5 CH
	1.96 (m)	38.4 CH		1.91 (m)	38.8 CH
	1.45, 0.97	25.01 CH2		1.46, 0.96	25.01 CH2
	1.11 (d,6.6)	14.8 CH3		1.11 (d, 6.6)	14.9 CH3
	0.92	12.4 CH3		0.91 m	12.4 CH3
	-	174.2 C		-	173.7 C
Phenoxazinone					
1		167.8 C			
2		101.8 C			
3		147.5 C			
4		179.9 C			
5		113.3 C			
6		145.5 C			
7		140.5 C			
8		127.7 C			
9	7.38 (d, 7.7)	130.4 CH			
10	7.66 (d, 7.7)	125.9 CH			
11		132.7 C			
12		129.4 C			
13		167.6 C			

## Data Availability

All data generated or analyzed for this study are included in the published article and Appendix A are available.

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
