# Peer review of "Antimicrobial Profile of Actinomycin D Analogs Secreted by Egyptian Desert Streptomyces sp. DH7"

_antibiotics, 2021, doi:10.3390/antibiotics10101264_

Round 1

Reviewer 1 Report

The paper by Hamin et al. provides an interesting  contribution in the area of research of new  compounds with antimicrobial activities and in particular shows the potential of bacteria isolated from desert environment and their activity against multiple drug resistant pathogens. However, there are several features of the manuscript that need to be improved, regarding both the English language  and, even more  importantly, the clarity in  the presentation of the data.

Line 20:  substitute “verified”   by  “displayed”

Line 28: analogs  instead of analog

Line 45  combat instead of compact.  Cancel “inevitability”

Line 47. belonging instead of belonged

Line 48. These Gram-positive  bacteria are  saprotrophic….

Line 102: Table 1:Test   should be Tested  isolate

Line 156: Biological evaluation of the active fraction of Streptomyces sp. DH7. “Biological evaluation” is vague. Evaluation of the antimicrobial activity  would be more appropriate. Moreover,  it is not clear what  the authors mean by “Active  fraction” can be a mixture containing various compounds  or  purified single compounds.  Interestingly   The text continues  describing the activities of 3 and 4  ( which are single compounds and not  fractions)

“Both compounds 3 and 4……”   these compounds are mentioned for  the first time, without any mention on their isolation from the active fractions. A hint to the appropriate section in the experimental part would be helpful.

Line 161: cancel nearly

Line 162: bioactive compound   which one?

Line:171: what is “biological compound”? maybe  bioactive metabolite ? If so, which one ?

Lines 177- 184 Fig 4 and Fig 5  active fractions ?(presumably) compounds 4 (  one purified compound)  ?

Line  191: Fig. 6  what is the purified active fraction? what  is the mic value referred to? Compound 3, compound 4,  the mixture of both ? Does it comprise  compounds 1 and 2 ?

.Line 199 :2.6. Identification of antimicrobial fractions

Rather than identification of the active fractions, it is the structural determination of the  compounds that had been isolated from the active fraction  however in the main these compounds are introduced  without any previous mention   of the procedure leading to their purification. Although it is described  in the experimental Section)   at  least a hint to the section  regarding the isolation of the bioactive compounds should be made.

Line:307  Our results confirmed that (32%) of the Actinobacteria isolates inhibited the growth of pathogenic multidrug-resistant S. aureus  Is it a new finding or a confirmation of a previous research ? that (32%) of the Actinobacteria isolates inhibited the growth of pathogenic multidrug-resistant S. aureus

Line 310 there is no reference  55

Line 325  Analysis of the bioactive fraction indicates that (Table 3) data confirmed that com-pound 2 is an ethyl derivative of 1. HSQC correlations confirmed single bond H-C connectives for compounds 1 and 2.

Replace  “Analysis” by NMR data of compounds 1 and2 instead of “bioactive fraction” also    “indicates … data confirmed “ ???

Line334  “had been disappeared” replace by  was not present

Line 341 Are there literature data on the antimicrobial properties of  these compounds and how do they compare with the results reported in this manuscript ?

Line 504:  Analysis     replace by analysis

Line 514: 4.9. Bioassay of the eluted fractions of Streptomyces sp. DH7

All fractions were eluted in an equal volume of sterilized DMSO ….

Where do these “fractions” come from? It is described in a subsequent section. It would be much easier to read the text if the description of the procedure  would reflect the logical  order of the experimental work. At least a reference to the appropriate section should be made.

Line 527: 4.10. Detection of minimum inhibitory concentrations of the active fraction of Streptomyces sp. DH7

The word detection should be substituted by determination

What is the active fraction =  an intermediate stage of the purification procedure or purified components . 

Line 552: Analogous comments …. What is the active fraction?

Line 577: VLC abbreviations  should be specified

Line 586: The paragraph describes the isolation of  compounds 3 and 4 The stepwise procedure used for their isolation  starts from fraction3 whose origin is not clearly described.

The yields of the pure compounds from the first eluate should be indicated.

Moreover,  it is elusive  how  were the compounds 1, 2, isolated, but no data relative  to the final purifications steps  are provided.

Author Response

We totally appreciate all the reviewer’s valuable comments and we have responded to them point by point.

All yellow highlights were paraphrased as requested by the reviewers.

Reviewers comments (1):

Line 20:  substitute “verified”  by  “displayed”

Author’s response: Done (ln 22)

Line 28: analogs instead of analog

Author’s response: Done  (ln 33)

Line 45  combat instead of compact.  Cancel “inevitability”

Author’s response: Done  ln(50)

Line 47. belonging instead of belonged

Author’s response: Done ln(50)

Line 48. These Gram-positive  bacteria are  saprotrophic….

Author’s response: Done ln (54)

Line 102: Table 1:Test   should be Tested  isolate

Author’s response: Done

Line 156: Biological evaluation of the active fraction of Streptomyces sp. DH7. “Biological evaluation” is vague. Evaluation of the antimicrobial activity  would be more appropriate. Moreover,  it is not clear what  the authors mean by “Active  fraction” can be a mixture containing various compounds  or  purified single compounds.  Interestingly   The text continues  describing the activities of 3 and 4  ( which are single compounds and not  fractions)

Author’s response: Thank you for this comment, we clarified that the active fraction is an intermediate fraction during the purification process, it was then further purified into 4 compounds. We clarified that in the text:( ln 187-ln189, ln 207, ln 214, we stated the definition of the active fraction, ln175-178, ln 587),

“Both compounds 3 and 4……”   these compounds are mentioned for the first time, without any mention on their isolation from the active fractions. A hint to the appropriate section in the experimental part would be helpful.

Author’s response: Thank you, it was clarified in (ln 187-ln 189)

Line 161: cancel nearly

Author’s response: Done

Line 162: bioactive compound   which one?

Author’s response: Thank you, it was clarified in (ln 178).

Line:171: what is “biological compound”? maybe bioactive metabolite ? If so, which one ?

Author’s response: Thank you, it was clarified in (ln 187-ln 189).

Lines 177- 184 Fig 4 and Fig 5  active fractions ?(presumably) compounds 4 (  one purified compound)  ?

Author’s response: Thank you, it was clarified in (ln 207, ln 2014) , legend of figure 5,6.

Line  191: Fig. 6  what is the purified active fraction? what  is the mic value referred to? Compound 3, compound 4,  the mixture of both ? Does it comprise  compounds 1 and 2 ?

Author’s response: Thank you, it was clarified in (ln 176-ln 177, ln 187-ln 189), it refers to an intermediate compound during the purification process and it is a mix of 4 compounds.

.Line 199 :2.6. Identification of antimicrobial fractions

Rather than identification of the active fractions, it is the structural determination of the  compounds that had been isolated from the active fraction  however in the main these compounds are introduced  without any previous mention   of the procedure leading to their purification. Although it is described  in the experimental Section)   at  least a hint to the section  regarding the isolation of the bioactive compounds should be made.

Author’s response: Thank you, it was clarified in (ln 228-ln 232).

Line:307  Our results confirmed that (32%) of the Actinobacteria isolates inhibited the growth of pathogenic multidrug-resistant S. aureus  Is it a new finding or a confirmation of a previous research ? that (32%) of the Actinobacteria isolates inhibited the growth of pathogenic multidrug-resistant S. aureus

 Author’s response: Thank you, it is a new finding in our study and agrees with previous reports in  (ln 350-ln 358-ln 360) ref. 35.

Line 310 there is no reference  55

Author’s response: Thank you, it is a S. aureus isolate number and it was adjusted in l(n  353). 

Line 325  Analysis of the bioactive fraction indicates that (Table 3) data confirmed that com-pound 2 is an ethyl derivative of 1. HSQC correlations confirmed single bond H-C connectives for compounds 1 and 2.

Replace  “Analysis” by NMR data of compounds 1 and2 instead of “bioactive fraction” also    “indicates … data confirmed “ ???

Author’s response: Done ln (369 - ln370).

Line334  “had been disappeared” replace by  was not present

 Author’s response: Done ln (379).

Line 341 Are there literature data on the antimicrobial properties of these compounds and how do they compare with the results reported in this manuscript?

Author’s response: Antimicrobial effect was reported for compounds 3 and 4 isolated our study (ln 406-408), ref 37. 

Line 504:  Analysis     replace by analysis

 Author’s response: Done ln (561).

Line 514: 4.9. Bioassay of the eluted fractions of Streptomyces sp. DH7

All fractions were eluted in an equal volume of sterilized DMSO ….

Where do these “fractions” come from? It is described in a subsequent section. It would be much easier to read the text if the description of the procedure would reflect the logical  order of the experimental work. At least a reference to the appropriate section should be made.

Author’s response: Thank you, this part was transferred in the logical order as suggested, item 4.13, (ln686-ln695).

Line 527: 4.10. Detection of minimum inhibitory concentrations of the active fraction of Streptomyces sp. DH7

The word detection should be substituted by determination

Author’s response: Done ln (611).

What is the active fraction = an intermediate stage of the purification procedure or purified components .

Author’s response: Done ln (561-562). 

Line 552: Analogous comments …. What is the active fraction?

What is the active fraction =  an intermediate stage of the purification procedure

Author’s response: we stated the definition of the active fraction, ln175-178, ln 587),

Line 577: VLC abbreviations  should be specified

Author’s response: it was clarified in the text as " vacuum liquid chromatography"(ln 652).

Line 586: The paragraph describes the isolation of  compounds 3 and 4 The stepwise procedure used for their isolation  starts from fraction3 whose origin is not clearly described.

The yields of the pure compounds from the first eluate should be indicated.

Moreover,  it is elusive  how  were the compounds 1, 2, isolated, but no data relative  to the final purifications steps  are provided.

Author’s response: the procedures for isolation of compounds 1-4 were clarified in details with the weights of the purified compounds in the material and methods part as required. 

Reviewer 2 Report

Dear Authors, 

The manuscript report the discovery of several new natural products from bacterium Streptomyces sp. DH7 through bioassay guided separation.

1) Abstract, please highlight the novelty of this manuscript, which natural products are new.

2) Page 7, line 201, "Compound 1", where 1 need to be bolded as compound 1. Please check all this mistake throughout the manuscript.

3) Figure 7, structures looked like being stretched. Please follow the chemdraw instruction from journal.

4) Figure 8, caption was incorrect, is suppose to be 1-4. But the structures 3-4 should fixed it position, because the top and right side of structures was out of frame, it was missing from figure.

5) Figure 9 can be omitted.

6) All the caption for structure's figure should remove "were prepared using ChemDraw".

7) Figure 10 is very blurred, should provide a better version.

8) Compounds 3-4 have two Proline in each structure. Proline can have two conformations cis and trans. By looking at Table 3, all four Proline are cis-Pro. Please include this information in the text. To determined this conformation, comparison of carbon chemical shifts at between carbon-beta and carbon-gamma. This information is provided in the following paper (as attached PDF file), and should be cited in the text.

J. Am. Chem. Soc. 2021, 143, 27, 10083–10087. DOI : 10.1021/jacs.1c05732  

9) The absolute configurations of 1-4 were not determined. Configuration of peptide 3-4 can be easily carry out by using Marfey's method.

Author Response

Reviewers comments (2):

  • Abstract, please highlight the novelty of this manuscript, which natural products are new.

Author’s response: the abstract was modified as required, compounds 1-4 are new natural products although they were previously synthesized.

  • Page 7, line 201, "Compound 1", where 1 need to be bolded as compound 1. Please check all this mistake throughout the manuscript.

Author’s response: all the compounds' numbers had been bolded as required

  • Figure 7, structures looked like being stretched. Please follow the chemdraw instruction from journal.

Author’s response: the structures were redrawn as required.

  • Figure 8, caption was incorrect, is suppose to be 1-4. But the structures 3-4should fixed it position, because the top and right side of structures was out of frame, it was missing from figure.

Author’s response: the caption had been corrected and the positions of the compounds had been fixed as required.

  • Figure 9 can be omitted.

Author’s response: yes, thank you, it had been omitted as required.

  • All the caption for structure's figure should remove "were prepared using ChemDraw".

Author’s response: the caption had been corrected as required.

  • Figure 10 is very blurred, should provide a better version.

Author’s response: figure 10 had been modified to be figure 9 after cancelation of fig. 9 and a better version had been added.

8) Compounds 3-4 have two Proline in each structure. Proline can have two conformations cis and trans. By looking at Table 3, all four Proline are cis-Pro. Please include this information in the text. To determined this conformation, comparison of carbon chemical shifts at between carbon-beta and carbon-gamma. This information is provided in the following paper (as attached PDF file), and should be cited in the text.

  1. Am. Chem. Soc.2021, 143, 27, 10083–10087. DOI : 10.1021/jacs.1c05732

 Author’s response: yes, thank you for this valuable comment, the reference had been included and the relevant information had been cited in the text.

  • The absolute configurations of 1-4were not determined. Configuration of peptide 3-4 can be easily carry out by using Marfey's method.

Author’s response: unfortunately, the purified amounts had been consumed in the biological assessments. However, the NMR data of compounds 3 and 4 are in excellent agreement for their very close derivatives cited in " Zhang, B.-z.; Wang, K.-r.; Yan, J.-x.; Zhang, W.; Song, J.-j.; Ni, J.-m.; Wang, R., In vitro and in vivo antitumor effects of novel actinomycin D analogs with amino acid substituted in the cyclic depsipeptides. Peptides 2010, 31, (4), 568-573." The absolute configuration could be matched with actinomycin D analouges cited in this reference no. (37).  

Reviewer 3 Report

Dear authors and Editor, this article confirms that, in nature, many organisms produce substances of high value. The article is interesting but needs to be made more captivating for readers. Therefore I ask  to improve  the text and the figures

Author Response

Dear authors and Editor, this article confirms that, in nature, many organisms produce substances of high value. The article is interesting but needs to be made more captivating for readers.Therefore I ask to improve the text and the figures.

Author’s response: All figures and text had been revised and modified as required by the reviewers. A better version of figure (10) had been added. Figure (9) was omitted as requested by reviewer (2).

Round 2

Reviewer 1 Report

The  authors inserted the suggested modifications, however there are still some points to be clarified.

Line 165 167

“Active fraction were purified into 4 compounds.  Replace by  “ four compounds could be isolated from the active fraction

 Line 602- 603: “After purification of the active fraction into 4 compounds, a volume of (20 µl) of each purified compound was added to sterile filter…”

 replace by : the  antimicrobial activity against MRSA strains of the four purified compounds was tested.

“     a volume of (20 µl) of each purified compound was added to sterile filter…”

“Only compounds 3 and 4 showed a high inhibitory effect against MRSA strains with (30-50 mm, respectively) diameter of inhibition zone (Figure 5-6).”

The quantity of each compound used for the test is not known. ( How much of each compound is dissolved in 20 µl  ?. Therefore the measurement of inhibition zones  does not provide reliable and significant information . The concentration of compound 3 and compound 4  should be indicated

Author Response

The authors inserted the suggested modifications, however there are still some points to be clarified.

Line 165 167

“Active fractions were purified into 4 compounds.  Replace by “four compounds could be isolated from the active fraction’

Authors response: Thank you, we have changed according to the reviewer comment, line 164.

 Line 602- 603: “After purification of the active fraction into 4 compounds, a volume of (20 µl) of each purified compound was added to sterile filter…”

 replace by : the  antimicrobial activity against MRSA strains of the four purified compounds was tested.

Authors response: Thank you, we have changed according to the reviewer comment, ln 603-604.

“     a volume of (20 µl) of each purified compound was added to sterile filter…”

“Only compounds 3 and 4 showed a high inhibitory effect against MRSA strains with (30-50 mm, respectively) diameter of inhibition zone (Figure 5-6).”

The quantity of each compound used for the test is not known. (How much of each compound is dissolved in 20 µl  ?. Therefore, the measurement of inhibition zones does not provide reliable and significant information. The concentration of compound 3 and compound 4 should be indicated.

Authors response: The concentration of each compound was added line 605-606.

Reviewer 2 Report

Dear Authors,

All the comments have been addressed, however:

I understood that all isolated compounds 1-4 were reported for the first time as natural products, but these structures of 1-4 have been published as synthetic products. Therefore, the stereochemistry of 1-4 should followed the synthetic products if their NMR data are identical.

Author Response

All the comments have been addressed, however:

I understood that all isolated compounds 1-4 were reported for the first time as natural products, but these structures of 1-4 have been published as synthetic products. Therefore, the stereochemistry of 1-4 should followed the synthetic products if their NMR data are identical.

Authors response:

Thank you, the stereochemistry of amino acid residues had been added as a footnote in Table 3 where all the residues are in the L-form except for D-valine. Also, this had been modified accordingly in Fig. 7.

Reviewer 3 Report

The scientific work can be published.

Author Response

The scientific work can be published.

Thank you for your comment.